# Effect of Carbon Nanoparticles on the Porous Texture of ι-Carrageenan-Based N-Doped Nanostructured Porous Carbons and Implications for Gas Phase Applications

Samantha K. Samaniego Andrade [1] , Alfréd Menyhárd [1], Szilvia Klébert [2] , Miklós Mohai [2] , Balázs Nagy [3] and Krisztina László [1,*]

[1] Department of Physical Chemistry and Materials Science, Faculty of Chemical Technology and Biotechnology, Budapest University of Technology and Economics, 1521 Budapest, Hungary; ssamaniegoandrade@edu.bme.hu (S.K.S.A.); menyhard.alfred@vbk.bme.hu (A.M.)

[2] Research Centre for Natural Sciences, Institute of Materials and Environmental Chemistry, Eötvös Loránd Research Network, Magyar Tudósok Körútja 2, 1117 Budapest, Hungary; klebert.szilvia@ttk.hu (S.K.); mohai.miklos@ttk.hu (M.M.)

[3] H-Ion Research, Development and Innovation Ltd., Konkoly-Thege út 29-33, 1121 Budapest, Hungary; balazs.nagy@hion.hu

\* Correspondence: laszlo.krisztina@vbk-bme.hu; Tel.: +36-309-781-999

**Abstract:** S and N double-doped high surface area biomass-derived carbons were obtained from marine biomass-derived ι-carrageenan. Adding carbon nanoparticles (CNPs), namely graphene oxide (GO) or carbon nanotubes (CNTs), in the early stage of the synthesis leads to a modified porous texture and surface chemistry. The porous textures were characterized by $N_2$ ($-196.15\ ^\circ C$) and $CO_2$ ($0\ ^\circ C$) isotherms. The best GO- and CNT-added carbons had an apparent surface area of 1780 m$^2$/g and 1170 m$^2$/g, respectively, compared to 1070 m$^2$/g for the CNP-free matrix. Analysis of the Raman spectra revealed that CNT was more efficient in introducing new defects than GO. Based on XPS, the carbon samples contain 2–4.5 at% nitrogen and 1.1 at% sulfur. The Dubinin–Radushkevich (DR) and Henry models were used to assess the strength of the interactions between various gases and the surface. The $N_2$/$H_2$ and $CO_2$/$CH_4$ selectivities were estimated with ideal adsorbed solution theory (IAST). While the CNPs, particularly GO, had a remarkable influence on the porous texture and affected the surface chemistry, their influence on the separation selectivity of these gases was more modest.

**Keywords:** carbon cryogel; heteroatoms; carbon nanotubes; graphene oxide; gas adsorption

## 1. Introduction

High surface area porous carbon materials have proved to be excellent media for gas phase-related applications, including separation and storage. High surface area and porosity are strongly associated. Carbon materials from renewable biomass provide a sustainable solution for the ever-increasing need for novel porous carbon precursors. While lignocellulosic biomass has long been used as a carbon precursor [1–6], crustacean waste, e.g., crab shells rich in chitosan [7], or plants of marine origin, e.g., seaweed [8], still represent an under-exploited area.

The versatility of porous carbon manufacturing allows the custom making of porous carbons for $CO_2$ enrichment from gas mixtures with $CO_2$ content of wide concentration range. Porous carbon materials are competitive adsorbents for the capture of low concentration (~400 ppm) $CO_2$ from air [9] as well as from flue gas or biogas [10,11] where the $CO_2$ concentration can be as high as 10–15% and 35–45%, respectively. In the latter case, biomethane (55–65%) is generally the main target of the separation.

Considering porous materials for adsorption-based gas storage or gas separation applications, pores having dimensions similar to the gas molecules of interest are the

most efficient. The diameters of $H_2$, $CO_2$, $O_2$, $N_2$ and $CH_4$ are 0.289, 0.33, 0.346, 0.362 and 0.38 nm, respectively [12]. The optimum ratio of pore size to adsorbate molecule size should be in the range of 1.7–3.0 [13], while in membranes, carbon molecular sieves with ultramicropores exhibit enhanced selectivity [14]. Although high micropore volume assures greater gas uptake [15,16], wider pores are not negligible either, as they enhance the dynamics of the transport processes. Physical or chemical activation methods are used to optimize the porous structure [3,17].

In addition to the physical interactions operating in the confinement of the pores, the surface chemistry of the carbon also plays a substantial role in surface interactions and thus in the selectivity. Heteroatoms, e.g., oxygen, nitrogen or sulfur that decorate the entrance and/or the pore walls, tune the hydrophobic/hydrophilic nature of the surface [18], modify the charge distribution of the neighboring carbon atoms [19–21], and thus provide a means to tune the selectivity of the porous carbon. As an example, Lewis basic functionalities (nitrogen and oxygen) present on the carbon surface may attract acidic $CO_2$ molecules, enhancing their uptake [22,23]. Sevilla et al. reported the synthesis of N-doping activated carbons with high surface area and large $CO_2$ capture capacity (7.4 mmol/g at 0 °C and 1 bar) [24]. Dual S and N doping results in a large number of carbon atom active sites through the redistribution of spin and charge densities as revealed by density functional theory (DFT) calculations [25].

Carbon nanoparticles (CNPs), like graphene derivatives or carbon nanotubes (CNTs) incorporated into the carbon matrices, may not only tune the porous texture but may create further defects that affect the selective interactions between the gas molecules and the surface. Alhwaige et al. reported that in chitosan–graphene oxide (GO) hybrid aerogels GO affects the apparent surface area and pore size distribution depending on the amount of GO added. An important result was an enhancement of the $CO_2$ capture capacity of the material with the addition of 20 wt% GO, due to increased surface area and pore volume. In addition, the chitosan-GO hybrid carbon showed enhanced attraction towards $CO_2$ molecules: -COOH, -OH, -$NH_2$, -$NO_2$, -$CH_3$ groups decorating the surface enhance the $CO_2$ adsorption by binding to $CO_2$ due to its quadrupole moment [7]. The computational studies of Bucior et al. found that the high separation selectivity and permeance of carbon nanotubes (CNTs) in the case of $H_2/CH_4$, $CO_2/CH_4$ mixtures can be attributed to $CH_4$ size exclusion [26]. CNP incorporation may also enhance the heat conductivity of the carbon matrix, thus contributing to faster removal of the heat escorting the adsorption process. Earlier, we studied the synthesis of a red algae-based heteroatom-doped carbon. Following the method of Li et al. [27], a highly porous carbon with an apparent surface area close to 1100 $m^2$/g was obtained. Due to the intrinsic S content of the marine biomass-based carrageenan and the N added with urea during the synthesis, a carbon with 5 at% O, 4.6 at% N and at 1% S was obtained. The gas separation and electric energy storage potential of this carbon was recently tested [28]. Here we report the effect of graphene oxide (GO) and multiwalled carbon nanotubes (MWCNT) on the porous texture and surface chemistry of that carrageenan–urea-based carbon matrix. The CNPs were added in the early stage, prior to the gelation of the parent matrix, and are thus expected to tune both the porous texture and the chemistry of the carbons obtained. After characterizing the morphology and surface chemistry, the $N_2/H_2$ and $CO_2/CH_4$ selectivity was estimated using ideal adsorbed solution theory (IAST) [29].

## 2. Materials and Methods

ι-carrageenan powder and urea pearls (98%) were purchased from Sigma Aldrich (Budapest, Hungary). The aqueous GO suspension (0.96 wt%) was prepared from natural graphite (Graphite Týn, Týn nad Vltavou, Czech Republic) following an improved Hummers' method [30,31]. The MWCNT was obtained from Chengdu Organic Chemicals Co., Ltd. (Chengdu, China). The as-received CNTs were oxidized in cc. $HNO_3$ (65%) for 3 h at 80 °C, washed with aqueous NaOH and reprotonated with 1.0 M HCl before use [32]. All other chemicals were used without further purification. Hydrogel matrices were ob-

tained by mixing 2 g urea and 2 g ι-carrageenan with a 100 mL aqueous CNP suspension (containing 50, 100 and 200 mg of the corresponding nanoparticles, respectively) at 80 °C. The freeze-dried polymer cryogels were pyrolyzed in a rotary quartz reactor at 700 °C (20 °C/min) in dry $N_2$ flow (25 L/h) for 1 h. The remaining inorganic impurities were removed by washing with 1.0 M HCl prior to annealing in Ar flow for 1 h at 1000 °C. The dry polymer and carbon cryogels were labelled PA and CA, respectively. The sample labels also refer to the incorporated CNP (as GO or CNT) and their incorporated mass (50, 100 or 200), respectively. CNP-free polymer and carbon gels were prepared for comparison [28].

*2.1. Characterization Methods*

Thermogravimetric analysis was performed on 2–10 mg polymer samples using a TGA 6 (Perkin Elmer, Waltham, MA, USA) thermogravimetric analyzer following a heating rate of 1.5 °C/min from room temperature up to 300 °C, and then 10 °C/min from 300 °C to 900 °C in nitrogen (20 mL/min).

Scanning electron micrographs were taken by a Zeiss Sigma 300 field emission scanning electron microscope (FESEM) (Carl Zeiss QEC GmbH, Oberkochen, Germany). Low temperature (−196.15 °C) nitrogen adsorption measurements were performed after 24 h degassing at 110 °C in a NOVA 2000 e (Quantachrome, Boynton Beach, FL, USA) instrument. The apparent surface area $S_{BET}$ was determined using the Brunauer–Emmett–Teller (BET) model [33]. The pore volume $V_{0.98}$ was estimated from the amount of vapor adsorbed at $p/p_0 = 0.98$, assuming that the adsorbed gas fills the corresponding pores as liquid. The Dubinin–Radushkevich (DR) model [34] was used to calculate the micropore volume $W_0$. The slope of the DR plots as well as the Henry constant (the initial slope of the isotherms) were used to characterize the interaction between the carbon surface and the adsorbate. The pore size distributions were computed using quenched solid density functional theory (QSDFT) for slit/cylindrical pore geometry [35]. Carbon dioxide adsorption was measured at 0 °C up to atmospheric pressure with an AUTOSORB-1 (Quantachrome, USA) analyzer. The pore size distribution in the ultramicropore range was derived by nonlinear density functional theory (NLDFT). Evaluation of the primary adsorption data was performed with the Quantachrome ASiQwin software (version 3.0). Raman spectra were obtained using a LabRAM (Horiba Jobin Yvon, Palaiseau, France) instrument. The laser source was a λ = 532 nm Nd-YAG (15 mW laser power at the focal point). A 0.6 OD filter was used to reduce the power of the beam. Parameter optimization and data analysis were performed by LabSpec 5 software.

X-ray photoelectron spectra were recorded on a Kratos XSAM 800 spectrometer operating in fixed analyzer transmission mode, using Mg $K_{\alpha1,2}$ (1253.6 eV) excitation. The analysis chamber pressure was below $1 \times 10^{-7}$ Pa. Survey spectra were recorded in the range 150–1300 eV in 0.5 eV steps. The photoelectron lines of C1s, O1s, N1s and S2p were measured in 0.1 eV steps with 1 s dwell time. The spectra were referenced to the energy of the C1s line of $sp^2$ type graphitic carbon, set at 284.3 ± 0.1 eV binding energy (BE). Peak decomposition was performed after Shirley-type background removal using a Gaussian–Lorentzian peak shape with 70:30 ratio as reported elsewhere [36]. Quantitative analysis, based on integrated peak intensity, was performed by the XPS MultiQuant program [37], applying the conventional infinitely thick layer model using the experimentally determined photoionization cross-section data of Evans et al. [38] and the asymmetry parameters of Reilman et al. [39]. Attenuated total reflectance Fourier transform infrared (FTIR-ATR) spectra were recorded on powdered carbons in the range 4000–400 $cm^{-1}$ at a resolution of 4 $cm^{-1}$ by 64 scans using a Tensor 27 (Bruker Optik GmbH, Leipzig, Germany) spectrophotometer equipped with a Platinum ATR unit A225. For the background signal, the measured medium was air. Since the absorption of the powders was very strong, a moderate polynomial baseline correction and smoothing were applied.

### 2.2. Adsorption with Probe Gases

Nitrogen ($-196.15$ °C), carbon dioxide and methane (both at 0 °C) isotherms were measured up to atmospheric pressure in a NOVA 2000 e (Quantachrome, USA) volumetric instrument. An Autosorb 1C (Quantachrome, Boynton Beach, FL, USA) volumetric instrument was used to perform hydrogen sorption experiments with high purity hydrogen (99.999%) at $-196.15$ °C. IAST [29] was used to assess the gas separation selectivity of the carbons studied.

## 3. Results and Discussion

### 3.1. Effect of the CNPs on the Morphology and Chemistry of the Samples

Figure 1 presents the thermogravimetric curve of the CNP-free and the 50 mg GO- and CNT-loaded polymer samples. The TG curves have multistep behavior with weight losses at similar temperatures, indicating that the thermal decomposition is governed by the pristine polymer. GO and CNT have little influence at these concentrations. The slightly different effect of the CNPs may stem from their dissimilar oxygen content (30 at% and 5 at% for GO and CNT, respectively).

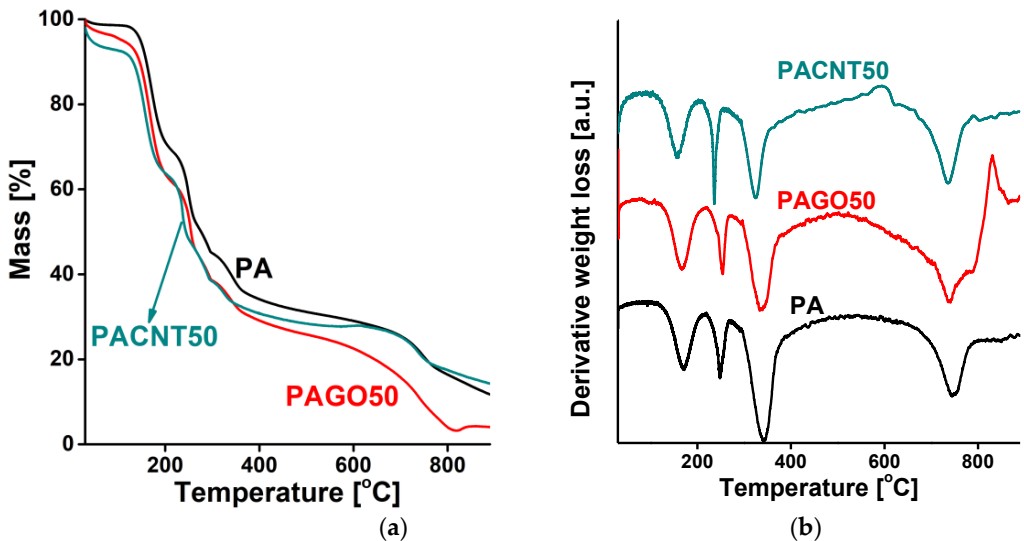

**Figure 1.** TG (**a**) and DTG (**b**) curves of the undoped polymer cryogel (PA), and the 50 mg CNP-doped polymer (PAGO50 and PACNT50) samples.

Figure 2 reveals the porous morphology of the carbon cryogels. Both CNPs seem well dispersed in the carbon matrix. The thermal treatment certainly reduces the GO that is incorporated into the carbon framework in the typical sheet-like form (Figure 2b). Similarly, CNTs added to the carbon cryogel kept their tubular shape (Figure 2c).

Low temperature $N_2$ adsorption isotherms and the respective pore size distribution curves for the GO- and CNT-doped carbons are shown in Figure 3. The numerical data deduced from these isotherms are presented in Table 1. According to the latest IUPAC classification, all the isotherms are a composite of Type II and IV, indicating the presence of micro-, meso- and macropores [40]. The H4 hysteresis loop having a sharp step-down around $p/p_0 = 0.45$ implies an interconnected pore network. Since the macropores are not totally filled with condensed nitrogen, the liquid equivalent volume $V_{0.98}$ was determined at $p/p_0 = 0.98$. According to Figure 3a, the surface-related properties of the carbon cryogel in the CAGO100 sample were most affected by addition of GO, both in the micro- and macropore regions, and 200 mg GO proved to be destructive due to the high amount of oxygen released by the GO during the heat treatment processes. The influence of CNT is more modest and slightly different. Adding only 50 mg CNT had an enhancing effect, but further addition of CNT gradually decreased both the apparent surface area and the pore volumes. The initial section of the $CO_2$ isotherms was utilized to reveal the

ultramicropore range, which is hardly accessible to the nitrogen molecules in cryogenic conditions. Figure 3b,d combines the pore size distributions in the ultramicropore and the micro-mesopore ranges. While in samples CAGO50 and CAGO100, GO increased the contribution of mesopores in the wider region (>10 nm), CNT had a widening effect in the narrow mesopore range. The distinctive impact of the GO and CNT may be related to their considerably different oxygen content as well as to their dissimilar geometry.

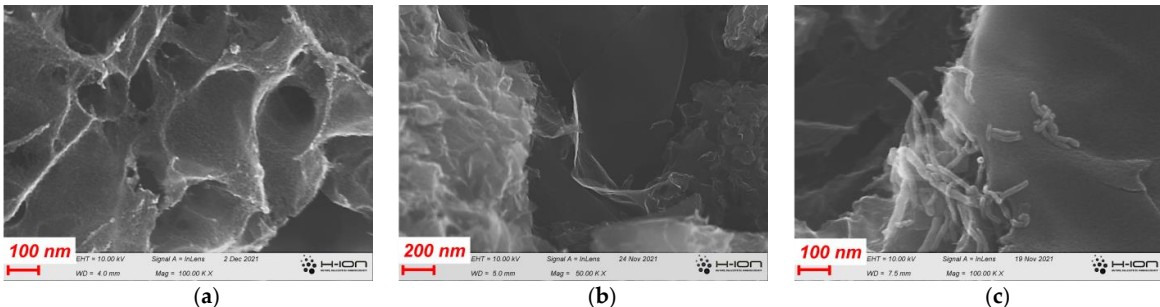

**Figure 2.** SEM image of the pristine carbon CA (**a**), GO-doped carbon CAGO50 (**b**) and CNT-doped carbon CACNT50 (**c**) samples.

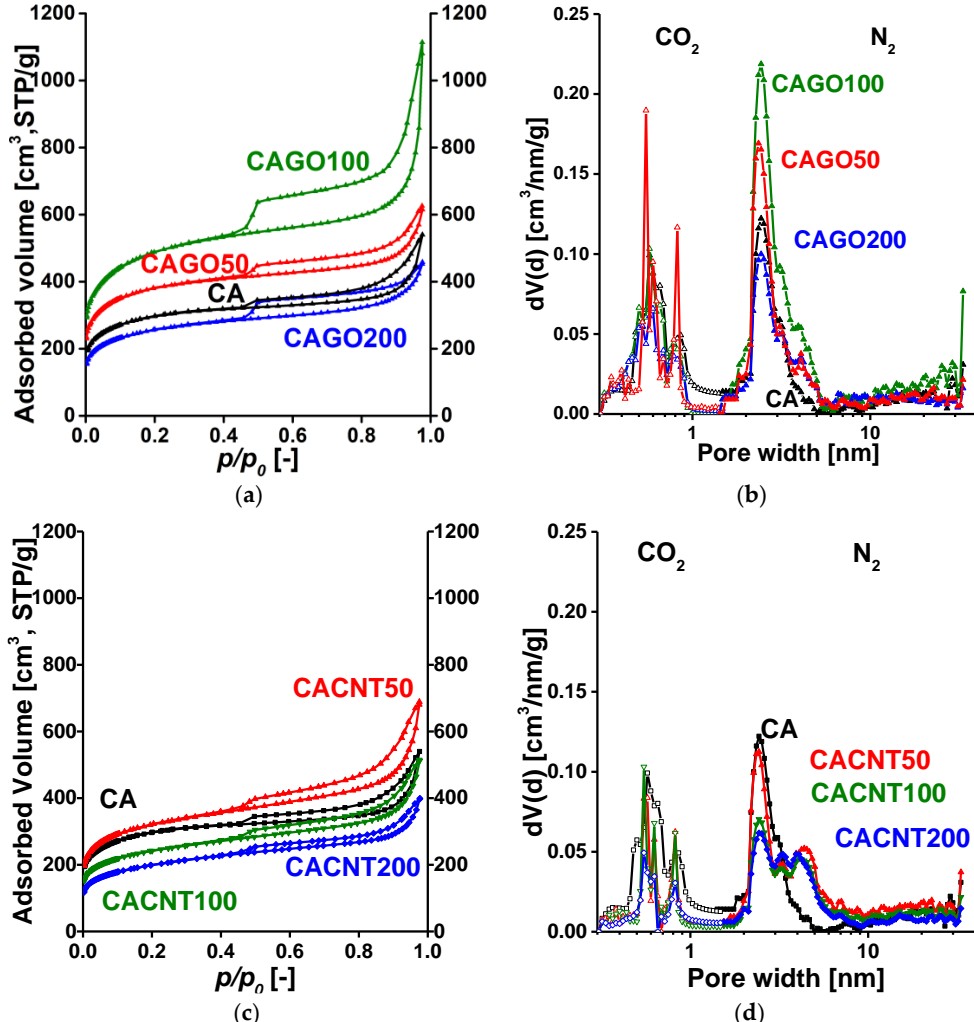

**Figure 3.** Low temperature N$_2$ adsorption/desorption isotherms of the GO- (**a**) and CNT- (**b**) doped carbon samples and their pore size distributions from 0 °C CO$_2$ adsorption (nonlinear density functional theory) and −196.15 °C N$_2$ adsorption data (quenched solid density functional theory, slit/cylindrical pores) (**c,d**).

**Table 1.** Data deduced from the low temperature $N_2$ and 0 °C $CO_2$ isotherms *.

| Sample | $S_{BET}$ | $V_{0.98}$ | $W_0$ | | $V_{meso}$ | $V_{umicro,DR}$ | $V_{umicro,DFT}$ |
|---|---|---|---|---|---|---|---|
| | | | from $N_2$ | | | from $CO_2$ | |
| | $m^2/g$ | $cm^3/g$ | $cm^3/g$ | % | $cm^3/g$ | $cm^3/g$ | $cm^3/g$ |
| CA | 1070 | 0.83 | 0.40 | 48 | 0.43 | 0.057 | 0.037 |
| CAGO50 | 1408 | 0.95 | 0.56 | 59 | 0.39 | 0.049 | 0.027 |
| CAGO100 | 1779 | 1.72 | 0.64 | 37 | 1.08 | 0.050 | 0.027 |
| CAGO200 | 933 | 0.71 | 0.34 | 48 | 0.37 | 0.040 | 0.022 |
| CACNT50 | 1169 | 1.07 | 0.43 | 40 | 0.64 | 0.032 | 0.018 |
| CACNT100 | 880 | 0.79 | 0.32 | 41 | 0.47 | 0.028 | 0.016 |
| CACNT200 | 727 | 0.62 | 0.26 | 42 | 0.36 | 0.020 | 0.013 |

\* Apparent surface area from BET model; $V_{0.98}$ is the liquid volume of the gas adsorbed at $p/p_0 = 0.98$; $W_0$ micropore volume from DR model; $V_{meso} = V_{0.98} - W_0$ mesopore volume; $V_{umicro,DR}$: ultramicropore volume (<0.7 nm) from DR model; $V_{umicro,DFT}$: ultramicropore volume (<0.7 nm) from quenched solid density functional theory (QSDFT, $CO_2$ adsorbed on carbon at 0 °C).

The Raman spectra in Figure 4 show the iconic D (~1350 cm$^{-1}$; defects, edges and disordered carbon sites) and G (~1580 cm$^{-1}$, $E_{2g}$ vibration of sp$^2$-hybridized graphitic carbon) band regions typical of carbon materials [41]. Addition of either CNP enhanced the formation of defects, as demonstrated by the increasing $I_D/I_G$ ratio with increasing amounts of additive. The effect is systematic and more pronounced in the CNT series.

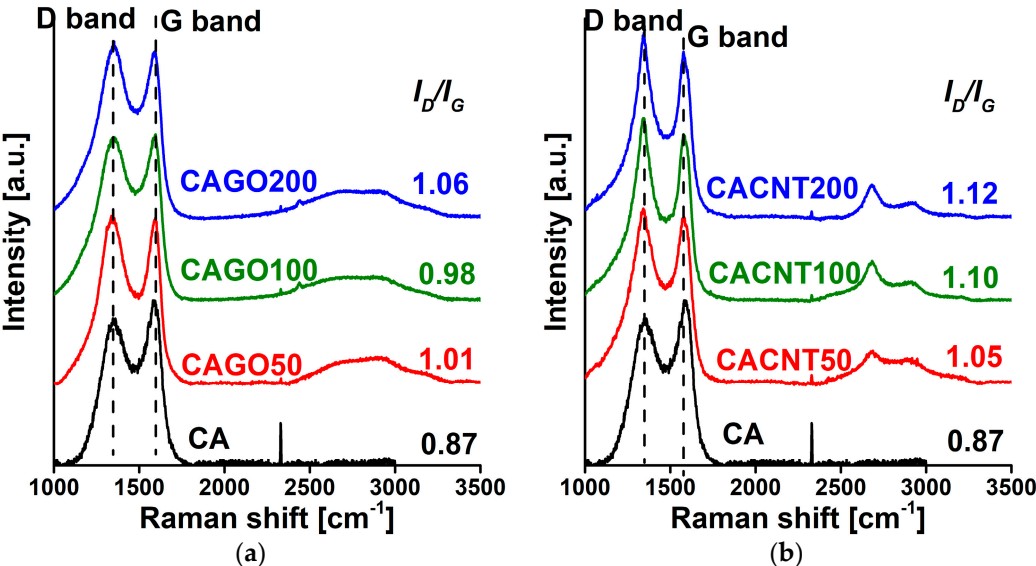

**Figure 4.** Raman spectra of the GO- (**a**) and CNT- (**b**) doped carbons.

The effect of the incorporated CNPs on the surface composition was studied by XPS (Table 2). In the CAGO samples, the C content increases in CAGO50 and then decreases for the two other GO concentrations, while for the CACNT samples the C content increases slightly but systematically. Regarding the nitrogen content, its concentration decreases in both sets of samples but most notably for the CACNT samples. In contrast, the S content does not significantly change among the samples. While the incorporation of GO—except the CAGO50 sample—has no effect on the total heteroatom ratio, CNT addition gradually decreases the (O + N + S)/C ratio.

**Table 2.** Surface composition (atomic %) measured by XPS.

| Sample | C | O | N | S | O/C | N/C | S/C | $\frac{O+N+S}{C}$ | S/N |
|---|---|---|---|---|---|---|---|---|---|
| CA | 90.6 | 3.3 | 5.1 | 1.0 | 0.036 | 0.056 | 0.011 | 0.104 | 0.196 |
| CAGO50 | 92.0 | 3.1 | 3.7 | 1.3 | 0.034 | 0.039 | 0.014 | 0.087 | 0.361 |
| CAGO100 | 90.7 | 4.1 | 4.1 | 1.2 | 0.045 | 0.045 | 0.013 | 0.104 | 0.293 |
| CAGO200 | 90.4 | 3.7 | 4.4 | 1.4 | 0.041 | 0.049 | 0.015 | 0.105 | 0.318 |
| GO | 67.4 | 32.1 | - | 0.5 | 0.476 | - | 0.007 | 0.484 | - |
| CACNT50 | 91.6 | 4.0 | 3.1 | 1.2 | 0.044 | 0.034 | 0.013 | 0.091 | 0.387 |
| CACNT100 | 91.8 | 3.7 | 3.4 | 1.1 | 0.040 | 0.037 | 0.012 | 0.089 | 0.323 |
| CACNT200 | 92.2 | 4.6 | 1.9 | 1.2 | 0.050 | 0.021 | 0.013 | 0.084 | 0.632 |
| CNT | 94.9 | 5.1 | - | - | 0.054 | - | - | 0.054 | - |

Figure 5 shows an example of the composite photoelectron lines (C1s, O1s, N1s and S2p) and their decomposition into the different chemical states. The binding energy ranges of the various states and their concentration are listed in Tables 3 and 4. Three different states of carbon, oxygen and nitrogen and two states of sulfur were distinguished in all the samples.

As XPS characterizes only the upper few nm of the samples, FTIR was also performed to reveal the composition in the deeper regions. The spectra for both sets of carbons are shown in Figure 6. The lines used for assignment are shown in Table 5 and the relative intensity ratios compared to the C=C signal are listed in Table 6. A systematic increase in the C=O/C=C and OH/C=C ratios was observed for both sets of samples, which may suggest that the thermal decomposition of both GO and the CNTs caused a relative increase in the oxygen functionalities C=O and OH with increasing CNP content. This agrees well with the chemical composition obtained from XPS.

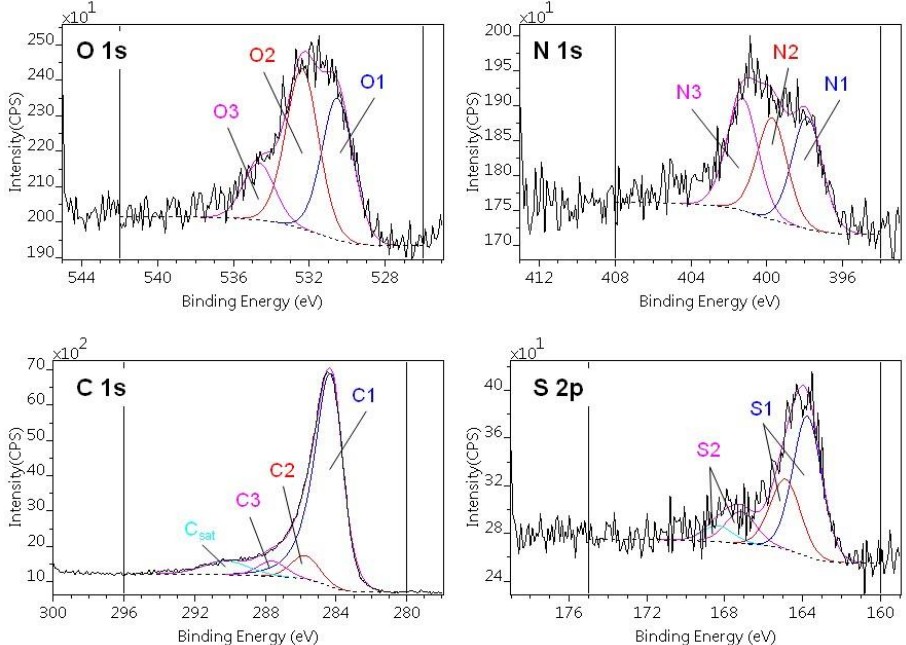

**Figure 5.** Decomposition of C1s, O1s, N1s and S2p regions of photoelectron spectra of the CACNT50 sample. Numerical data are given in Tables 3 and 4.

**Table 3.** Decomposition of C1s and O1s regions of photoelectron spectra: binding energy ranges, chemical state assignations and surface compositions (atomic %) [36].

| | C1s | | | O1s | | |
|---|---|---|---|---|---|---|
| | **C1** | **C2** | **C3** | **O1** | **O2** | **O3** |
| **Chemical state** | $sp^2$ C=C | C–O<br>C–N<br>C–S | C=O<br>O–C–O<br>N–C–O | S–O | C–O–C<br>C–OH<br>C=O | OC–O–CO<br>($H_2O$) |
| **Binding energy [eV]** | 284.3–284.4 | 285.7–285.8 | 287.5–287.9 | 530.2–530.6 | 532.1–532.5 | 533.9–534.3 |
| CA | 74.0 | 10.9 | 5.4 | 1.5 | 1.7 | n.d. |
| CAGO50 | 78.8 | 7.4 | 5.5 | 1.9 | 1.3 | n.d. |
| CAGO100 | 74.7 | 11.0 | 4.8 | 1.8 | 1.7 | 0.7 |
| CAGO200 | 75.9 | 9.4 | 4.8 | 1.8 | 1.6 | 0.5 |
| CACNT50 | 78.6 | 7.8 | 5.1 | 1.6 | 1.9 | 0.7 |
| CACNT100 | 80.5 | 7.4 | 4.5 | 1.3 | 2.0 | 0.7 |
| CACNT200 | 75.7 | 10.8 | 5.5 | 1.7 | 2.3 | 0.9 |

**Table 4.** Decomposition of N1s and S2p regions of photoelectron spectra: binding energy ranges, chemical state assignations and surface compositions (atomic %) [36].

| | N1s | | | S2p | |
|---|---|---|---|---|---|
| | **N1** | **N2** | **N3** | **S1** | **S2** |
| Chemical state | C–N | OO–C–N | C–N$^+$ | C–S | C–SO$_3$ |
| Binding energy [eV] | 397.8–398.0 | 400.4–400.5 | 402.4–402.7 | 164.9–165.0 | 168.3–168.6 |
| CA | 2.3 | 2.3 | 0.8 | 0.9 | 0.2 |
| CAGO50 | 1.6 | 1.7 | 0.6 | 1.2 | n.d. |
| CAGO100 | 1.9 | 1.9 | 0.4 | 1.0 | 0.2 |
| CAGO200 | 2.0 | 2.0 | 0.7 | 1.1 | 0.3 |
| CACNT50 | 1.1 | 1.0 | 1.1 | 1.0 | 0.2 |
| CACNT100 | 0.9 | 0.7 | 0.9 | 1.0 | 0.2 |
| CACNT200 | 0.6 | 0.5 | 0.8 | 0.9 | 0.4 |

**Table 5.** Assignation of the FTIR peaks [42].

| Wavenumber [cm$^{-1}$] | Assignation |
|---|---|
| 1750–1705 | aromatic (1730–1705) and aliphatic (1750–1730) C=O stretching |
| 1600–1400 | C=C bond stretching in the aromatic ring |
| 1350 ± 50 | OH in-plane bending of phenol and alcohol groups |
| 1260–1200 | C–O stretching in phenols |
| 1060–1035 | C-O stretching in noncyclic acid anhydrides |

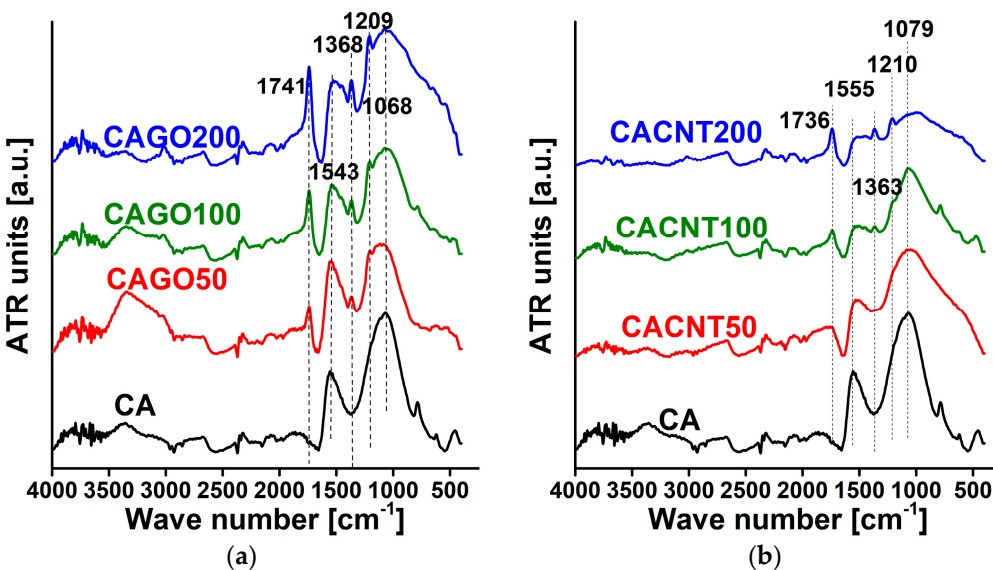

**Figure 6.** FTIR spectra of GO-doped carbons (**a**) and CNT-doped carbons (**b**).

**Table 6.** Intensity ratios of FTIR signal.

| Sample | C=O/C=C | OH/C=C | C-O(H)/C=C | C-O-C/C=C |
|---|---|---|---|---|
| CA | 0.19 | 0.50 | 1.33 | 1.69 |
| CAGO50 | 0.53 | 0.62 | 1.10 | 1.17 |
| CAGO100 | 0.93 | 0.78 | 1.31 | 1.51 |
| CAGO200 | 1.17 | 1.02 | 1.55 | 1.60 |
| CACNT50 | 0.53 | 0.81 | 1.50 | 1.86 |
| CACNT100 | 0.91 | 1.00 | 1.73 | 2.68 |
| CACNT200 | 1.33 | 1.33 | 1.72 | 1.89 |

*3.2. Gas Storage and Separation Results*

The potential of these carbons in gas separation was based on single gas adsorption measurements. The $N_2$, $CO_2$, $H_2$ and $CH_4$ adsorption isotherms of the various GO-doped carbon samples are presented in Figure 7. For easier comparison, only the adsorption branches are plotted and the gas uptakes are expressed in mmol/g. It should be noted that the effect of the incorporated GO varies from adsorbate to adsorbate. Incorporation of GO in the early stage of the synthesis affects not only the gel formation, but even more the porous texture and the surface chemistry in a sophisticated way. On comparing the $N_2$ and $H_2$ uptakes at $-196.15$ °C, it is clear that all the samples adsorb significantly more $N_2$ than $H_2$, as the boiling point of $H_2$, $-252.9$ °C, is much lower than the temperature of the uptake measurements. The almost tenfold difference indicates the potential of these carbon samples for $N_2/H_2$ separation at the temperature of these measurements. It is known that narrow micropores decorated with oxygen and nitrogen functional groups enhance $CO_2$ uptake [22]. The isotherms of $CO_2$ and $CH_4$ adsorption (0 °C) show that all samples adsorb more than twice as much $CO_2$ as $CH_4$ and CAGO50 attains the highest uptake for both gases at atmospheric pressure. This could indicate the potential of the GO-doped samples for $CO_2/CH_4$ separation.

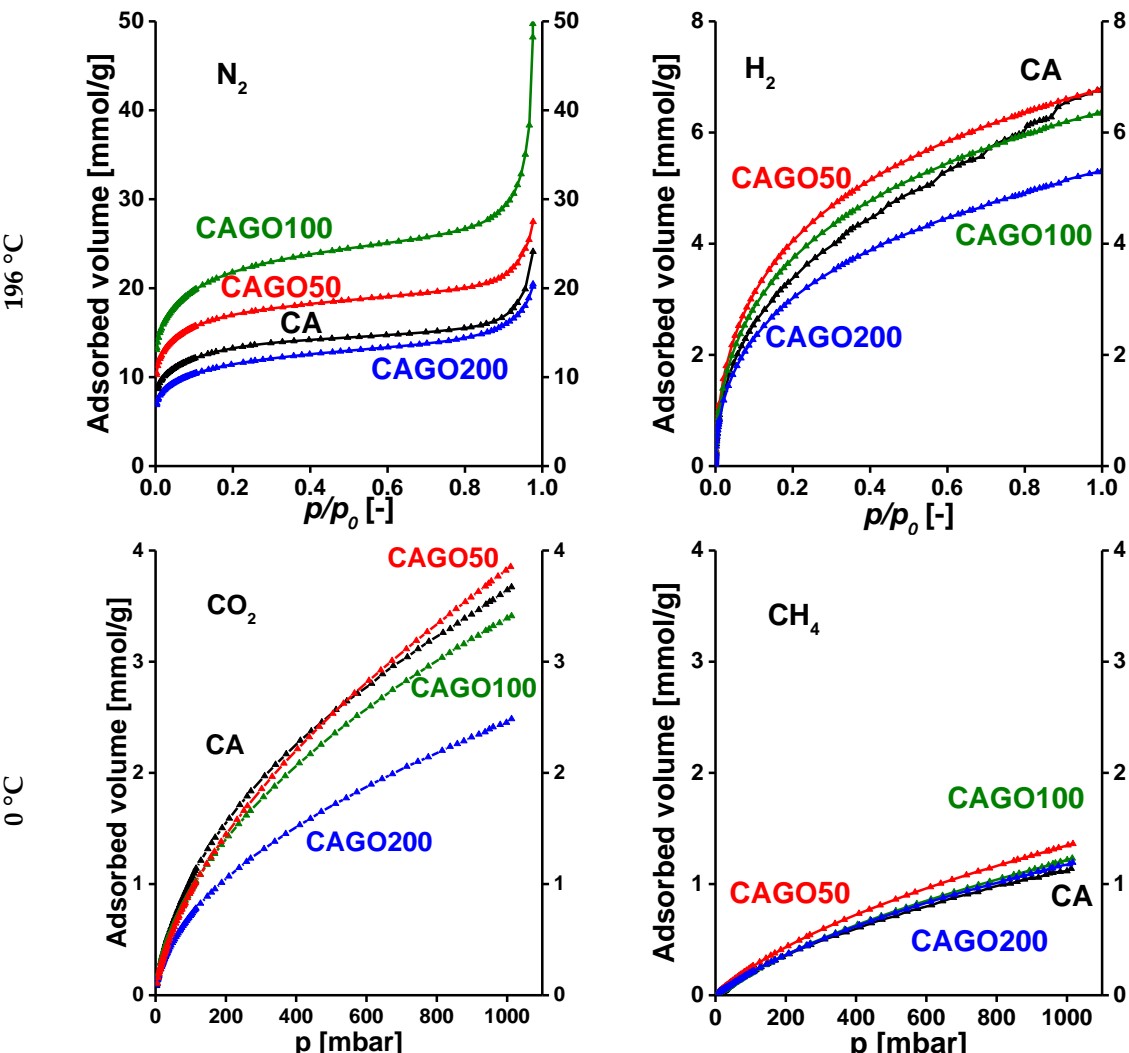

**Figure 7.** Adsorption isotherms of GO-doped carbon cryogels. $N_2$ and $H_2$ were measured at $-196.15\ °C$, while $CO_2$ and $CH_4$ isotherms were measured at $0\ °C$.

Similarly, Figure 8 presents the $N_2$, $H_2$, $CO_2$ and $CH_4$ adsorption isotherms of the annealed CNT-doped carbons. Here, the sequence of the overall uptakes is similar at the two temperatures, respectively. Generally, the CNT-doped samples display a poorer adsorption performance (proportional to the incorporated CNT) than the undoped CA carbon. Only sample CACNT50 has somewhat higher uptakes for both gases measured at $-196.15\ °C$.

In order to compare the interaction between the carbon surface and the probe gases, we used the corresponding fitting parameters of the DR and the Henry models. On the one hand, the slope of the linearized DR plot was used as follows,

$$\ln W = \ln W_0 - \left(\frac{RT}{E}\right)^2 \ln^2 \frac{p_0}{p} \tag{1}$$

where $W$ is the actual filling of the micropore volume $W_0$, $E$ is the characteristic energy of the given system, $p$ is the equilibrium pressure and $p_0$ is the saturation pressure of the probe gas at the temperature $T$ of the measurement. On the other hand, the Henry constant $K_H$ was determined, where the initial section of the isotherm was fitted to the Henry model as follows,

$$n = K_H \frac{p}{p_0} \tag{2}$$

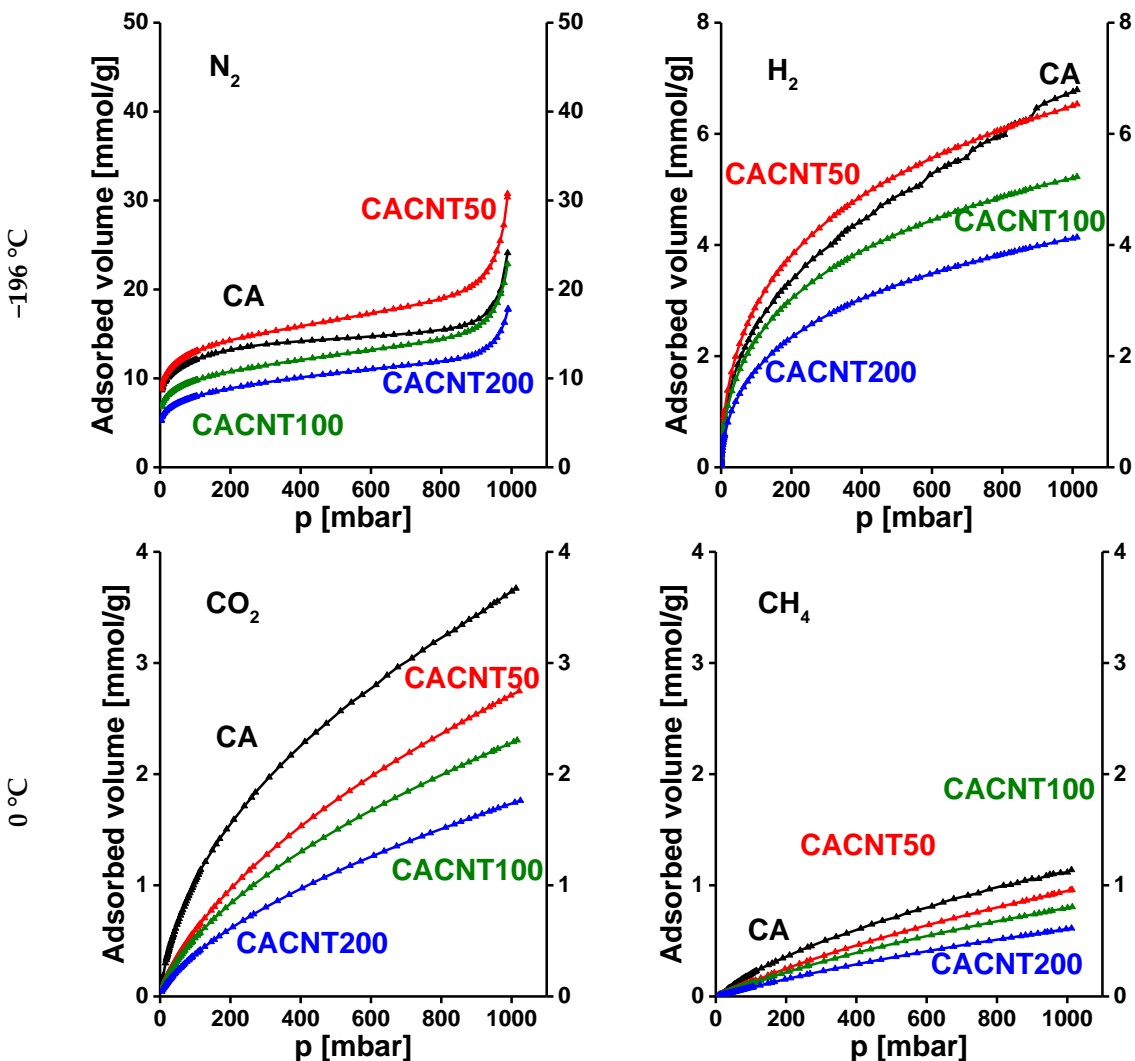

**Figure 8.** Adsorption isotherms of CNT-doped carbon cryogels. $N_2$ and $H_2$ were measured at $-196.15\ °C$, while $CO_2$ and $CH_4$ isotherms were measured at $0\ °C$.

Here, *n* is the amount of gas adsorbed (mmol/g) at the corresponding relative pressure. The numerical data are given in Table 7.

**Table 7.** Interaction-related parameters and their ratios from the probe gas isotherms in Figures 7 and 8.

| | | | CA | CAGO50 | CAGO100 | CAGO200 | CACNT50 | CACNT100 | CACNT200 |
|---|---|---|---|---|---|---|---|---|---|
| $-196\ °C$ | $N_2$ | $\left(\frac{RT}{E}\right)^2$ | 0.0236 | 0.0254 | 0.0271 | 0.0278 | 0.0297 | 0.0292 | 0.0298 |
| | | $K_H$ * | 0.151 | 0.284 | 0.338 | 0.156 | 0.213 | 0.152 | 0.140 |
| | $H_2$ | $\left(\frac{RT}{E}\right)^2$ | 0.0978 | 0.106 | 0.108 | 0.0988 | 0.0842 | 0.116 | 0.0870 |
| | | $K_H$ | 0.543 | 0.661 | 0.541 | 0.437 | 0.430 | 0.329 | 0.241 |
| $0\ °C$ | $CO_2$ | $\left(\frac{RT}{E}\right)^2$ | 0.212 | 0.196 | 0.182 | 0.163 | 0.197 | 0.198 | 0.201 |
| | | $K_H$ | 0.0186 | 0.0166 | 0.0190 | 0.0148 | 0.00950 | 0.00890 | 0.00620 |
| | $CH_4$ | $\left(\frac{RT}{E}\right)^2$ | 0.551 | 0.239 | 0.195 | 0.251 | 0.236 | 0.239 | 0.220 |
| | | $K_H$ | 0.00210 | 0.00300 | 0.00210 | 0.00240 | 0.00140 | 0.00120 | 0.000900 |

\* Expressed in mmol/(g·mbar).

The relative interactions were characterized by the ratio of the corresponding characteristic energies derived from the DR slopes and by the ratio of the $K_H$ values (Figures 9 and 10). Comparison of the DR interactions reveals that the energy ratio governing the $N_2/H_2$ separation is higher than that of the $CO_2/CH_4$ separation. The addition of the nanoparticles does not enhance the energy ratios, but leads to a ca. 25% loss in the case of $CO_2/CH_4$ separation.

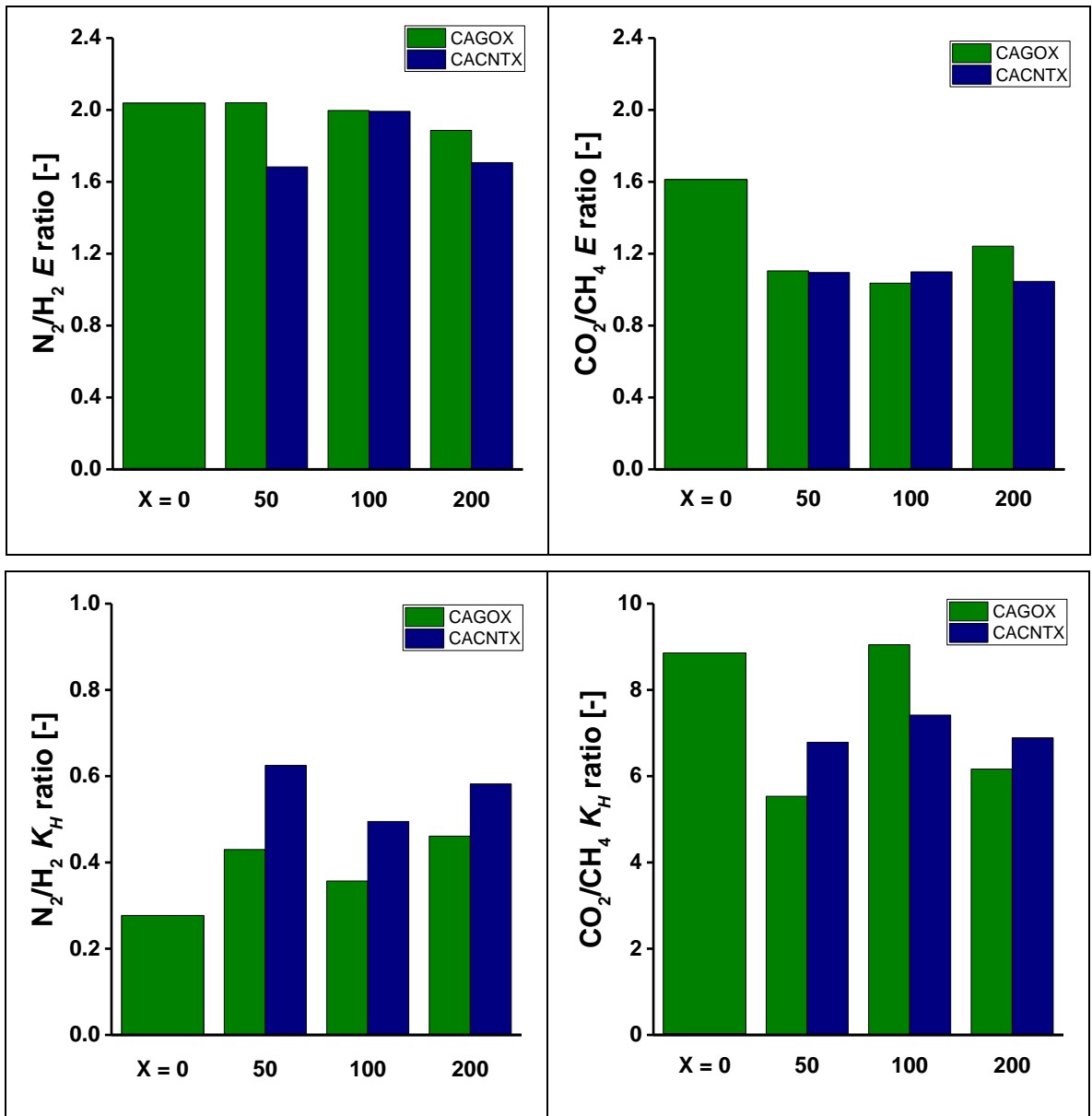

**Figure 9.** Comparison of characteristic parameters of gas/surface interactions.

The trend revealed of the Henry constant ratios are slightly different, as $K_H$ is a more complex parameter, characterizing the distribution rather than the interaction itself. The Henry ratios are about ten times higher in the $CO_2/CH_4$ separation than in that of the $N_2/H_2$. While the CNPs, particularly the nanotubes, have an enhancing effect for the latter, they do not improve this ratio for $CO_2/CH_4$.

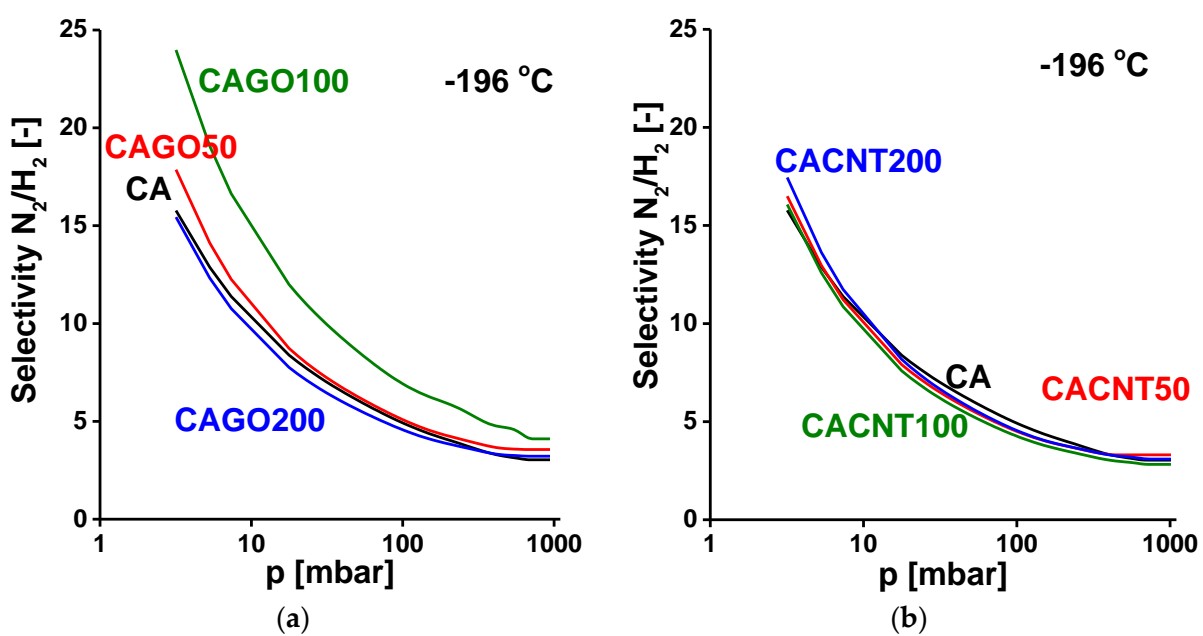

**Figure 10.** $N_2/H_2$ selectivity curves of GO-doped carbons (**a**) and CNT-doped carbons (**b**).

As in the work of Kamran et al. [43], IAST [29] was also used with the fitted adsorption data to determine and compare the $N_2/H_2$ and $CO_2/CH_4$ selectivity of the GO- and CNT-doped carbon samples. The $N_2$ adsorption isotherms were fitted to polynomial curves, while the $CO_2$, $CH_4$ and $H_2$ adsorption isotherms were fitted to the single-site Langmuir–Freundlich model [44,45] as follows,

$$n = \frac{n_{sat}Kp^m}{1 + Kp^m} \tag{3}$$

where $n$ is the adsorbed quantity at equilibrium pressure $p$, $n_{sat}$ is the saturation capacity, $K$ is the equilibrium constant of the Langmuir model and $m$ (>1) is the Freundlich exponent. The selectivity curves for the $N_2/H_2$ system are shown in Figure 10, and those corresponding to the $CO_2/CH_4$ system are shown in Figure 11. In all the cases studied, the selectivity gradually reduces with increasing pressure. As expected, all the carbons adsorb nitrogen preferentially to hydrogen at −196 °C. The GO-doped carbons display a selectivity that is also influenced by the added GO. The CAGO100 sample is significantly better than the pristine CA carbon over the whole pressure range. For the CNT-doped carbons, all the selectivity curves lie close to that of the CA carbon, which indicates that the inclusion of CNTs did not affect the $N_2/H_2$ selectivity.

In $CO_2/CH_4$ separation at 0 °C (Figure 10), all the CNP-incorporated samples performed less well than the undoped CA carbon. Only sample CAGO100 reached a selectivity similar to CA in the lower pressure range. Apart from this case, the CNT-doped samples exhibited better selectivity than the GO family.

The trends observed from the three sets of comparisons confirm that relative adsorption and thus selectivity are a trade-off between multiple kinetics and diffusion-controlled processes (neither being independent of pore morphology and surface interactions). The most complex information is delivered by selectivity curves, but even they are only estimates that ignore any time-dependent and technical aspect of the separation.

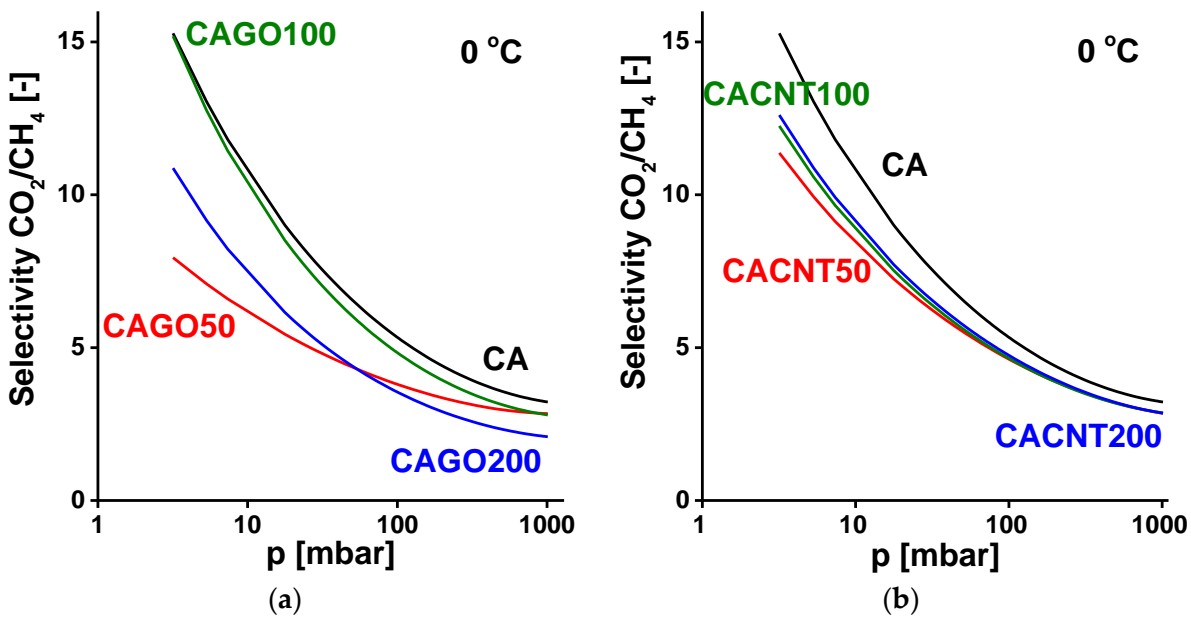

**Figure 11.** $CO_2/CH_4$ selectivity curves of GO-doped carbons (**a**) and CNT-doped carbons (**b**).

## 4. Conclusions

N, S double-doped porous carbon samples incorporating GO and CNT were successfully obtained from ι-carrageenan by applying urea as a nitrogen source during the synthesis. The carbon nanoparticles conserved their characteristic shape and were distributed homogeneously in the precursor gel. Their addition yielded a modified porous texture, particularly with GO with its significantly higher oxygen content. The highest apparent surface area and pore volume was found in the case of GO (sample CAGO100 possesses 1780 $m^2/g$ and 1.72 $cm^3/g$, respectively), while with CNT a more modest improvement was achieved (CACNT50: 1169 $m^2/g$ and 1.07 $cm^3/g$, respectively). The overall heteroatom concentration decorating the surface was slightly higher after the GO incorporation, as the nitrogen content was conserved more efficiently. The CNPs introduced further disorder into the matrix, as revealed by Raman spectroscopy. We can conclude that CNP, particularly GO incorporation, is more efficient in tuning the porous texture than the surface chemistry. Nitrogen and hydrogen isotherms measured at $-196.15\ °C$ and $CO_2$ and $CH_4$ isotherms at $0\ °C$ were used to assess the effect of the CNPs on the selectivity of these carbons. The three different approaches applied to assess selectivity show different trends. The most complex method, IAST, revealed that incorporation of CNPs reduced rather than improved selectivity compared to neat CA itself in all the cases. The only exception was CAGO100, the best of the GO samples, which displayed enhanced selectivity in $N_2/H_2$ separation and a performance similar to CA for $CO_2/CH_4$. The estimates given here are based on the individual equilibrium isotherms and thus exclude any kinetic or diffusion-related mechanisms.

**Author Contributions:** Conceptualization: K.L.; methodology: M.M. and A.M.; investigation: S.K.S.A., M.M., S.K. and B.N.; resources: K.L.; writing—original draft preparation: K.L. and S.K.S.A.; writing—review and editing: K.L. All authors have read and agreed to the published version of the manuscript.

**Funding:** Financial support from the Hungarian Scientific Research Funds OTKA K128410 is acknowledged. The research is part of project no. BME-NVA-02, implemented with the support of the Ministry of Innovation and Technology of Hungary from the National Research, Development and Innovation Fund, and financed under the TKP2021 funding scheme. SKSA is grateful to the Stipendium Hungaricum scholarship program of the Hungarian Government.

**Data Availability Statement:** Data will be made available on request.

**Acknowledgments:** We extend our warm thanks to A. Farkas for his invaluable assistance in Raman spectroscopy and to G. Bosznai for his technical assistance.

**Conflicts of Interest:** The authors declare no conflict of interest.

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
