# Peer review of "Effect of Carbon Nanoparticles on the Porous Texture of ι-Carrageenan-Based N-Doped Nanostructured Porous Carbons and Implications for Gas Phase Applications"

_carbon, 2023_

Round 1

Reviewer 1 Report

The work of Andrade et al. assess the properties of porus carbon compounds obtained by mixing carrageenan and carbon nanotubes or graphene oxides. The work does not present any breakthrough findings or advance particularly the field. However, in my opinion it meets the standard for publication in peer-reviewed journals. I have suggested below some comments for the authors to further improve their manuscript.

The title of the paper needs to be simplified for clarity

The introduction seems to blur the topic of greenhouse/flue gases capture separation with that of carbon compounds at high surface area. It is not clear how the two relates to each other. For example

-          Line 32. How gas separation is related to carbon storage mentioned in line 31?

-          Line 34. Novel carbon precursor for which application? The flue gas separation? How? In which industrial settings?

-          Line 47. Carbon molecular sieve….how are these to be deployed? In membranes? In energy-producing plants? In flue gas producing plants?

-          Line 62. What is the CO2 produced in the synthesis of CNP/CNT?

-          Line 69. Are these properties due to a synergistic effect between the aerogel and GO or simply the effect of the GO? Then it is not clear why one would need the aerogel support if GO can be used directly. Again, the authors do not explain how these carbon materials would be used in industrial settings

What is the role of S and N in line 74 to 86?

Line 93. MWCNT were oxidized. How?

Line 115. A few words on why this specific model was chosen could be said. BJH is usually more common.

Line 145. This N isotherm is the same used for the BET determination?

Table 1. Did you provide an explanation why CAGO200 has a decreased surface area?

Figure 2. Are these images before pyrolysis correct?

Line 263 onwards. This does not really explain the role of the heteroatom in gas absorption

Line45. Atomic radii are usually given in Angstrom

Line 46 Range is misspelled as rage

The manuscript reads well. I still recommend to re-check the manuscript for flow and spelling. No professional check is needed in my opinion

Author Response

Please, see the attachments

Reviewer 2 Report

The manuscript reports the incorporation of two types of nano-structured carbon, namely graphene oxide (GO) and carbon nanotubes (CNTs), into a N- and S-doped cryo-gel obtained from a marine biomass-related precursor. The authors report how the porosity and the surface chemistry are affected by both the types of additives, as a function of concentration. The characterization of the various materials uses a wide variety of techniques that are presented clearly. I would propose few minor corrections before publication:

-        Fig. 1a: all TGA curves should start from 100% weight. By the way, on the y axis what is plotted is the weight, not the weight loss (panel b should be modified accordingly).

-        Resolution of most figures and plots should be improved. In particular, SEM images in Fig. 2 are not readable at all, including their respective scale bars. The aim of these images is to show incorporation of CNPs within the gel and its homogeneity, if detectable with the used SEM. In the caption the particular sample should be mentioned

The quality of English is quite good, but it should be checked for small typos.
